# Beyond the Standard Model New Physics Searches with SBND †

**Supraja Balasubramanian on behalf of the SBND Collaboration**

Fermi National Accelerator Laboratory, Batavia, IL 60510-5011, USA; suprajab@fnal.gov

† Presented at the 23rd International Workshop on Neutrinos from Accelerators, Salt Lake City, UT, USA, 30–31 July 2022.

**Abstract:** SBND (Short-Baseline Near Detector) is a 112-ton liquid argon time projection chamber located on the Booster Neutrino Beam at Fermi National Accelerator Laboratory, and is the near detector of the Short-Baseline Neutrino program. The primary goals of SBND are to provide flux constraints for sterile neutrino searches, conduct world-leading neutrino cross-section measurements on argon, and perform Beyond the Standard Model (BSM) new physics searches with its high-precision particle identification capabilities. SBND's prospects and tools for detecting a variety of BSM phenomena produced in a neutrino beam, such as sub-GeV dark matter, dark neutrinos, heavy neutral leptons and millicharged particles, are discussed.

**Keywords:** neutrinos; Beyond the Standard Model; liquid argon TPCs; SBN; Fermilab; neutrino oscillations





## 1. Introduction

The Short-Baseline Neutrino (SBN) [1] program at Fermilab consists of three detectors with liquid Argon time projection chamber (LArTPC) technology on the Booster Neutrino Beam (BNB): SBND (110 m from the target; start of operations expected in early 2024), MicroBooNE [2] (470 m from the target; data-taking from 2015–2021) and ICARUS [3] (600 m from the target; currently operating). MicroBooNE was built primarily to investigate the "low-energy excess" of electromagnetic events observed by the MiniBooNE experiment [4], which could have been a hint of sterile neutrinos—MicroBooNE did not observe a similar excess [5,6], but this does not exclude the possibility of sterile neutrinos or alternative Beyond the Standard Model (BSM) explanations for the MiniBooNE excess. Moving forward, the SBN physics program (now with a large-mass far detector, ICARUS, and a high-statistics near detector, SBND) aims to substantially improve the global dataset of sterile neutrinos through neutrino oscillation searches in the electron neutrino appearance, muon neutrino disappearance and electron neutrino disappearance channels on the Booster Neutrino Beam.

The BNB is made by training an 8 GeV high-intensity proton beam on a beryllium target, producing a predominantly muon neutrino beam (see Figure 1). SBND's proximity to the beam target positions allows it to collect the largest statistics of neutrino interactions on argon before the DUNE experiment comes online—SBND is projected to take $10–18 \times 10^{20}$ protons-on-target (POT) of data over a 3–4 year run. This is crucial for constraining both flux and cross-section systematic uncertainties for neutrino oscillation experiments. In addition, the high-intensity proton beam can be a source of a variety of theoretically motivated BSM new physics phenomena, which would subsequently interact in SBND and produce detectable signatures.

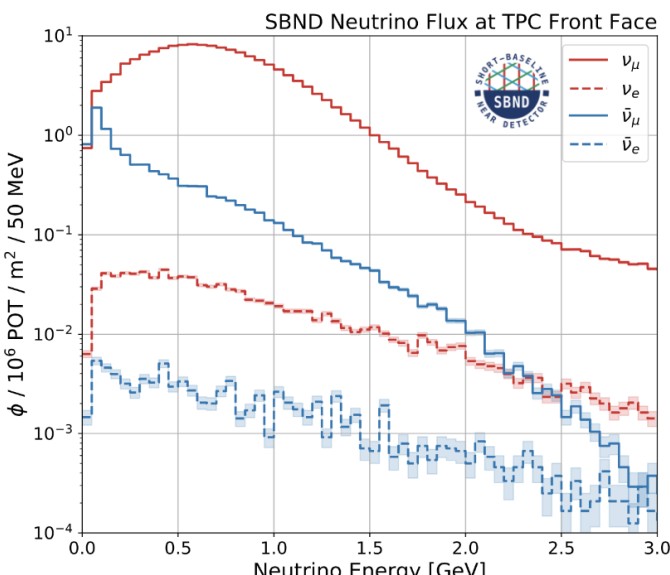

**Figure 1.** Booster Neutrino Beam flux at SBND.

## 2. The SBND Experiment

SBND is a liquid argon time projection chamber that consists of different detector subsystems, each of which allows it to provide full 3D event reconstructions of particle interactions with mm-scale precision, nanosecond-scale timing resolution, and low particle detection thresholds.

### 2.1. Time Projection Chamber

The main detector volume (4 m × 4 m × 5 m) is divided into two liquid argon time projection chambers with a cathode plane in the middle and anode wire planes on either side (see Figure 2). Ionization electrons produced in an interaction are drifted along each outgoing particle's trajectory using an electric field of 500 V/cm and collected at three layers of anode plane wires, which are spaced 3 mm apart—this allows for mm-scale position resolution.

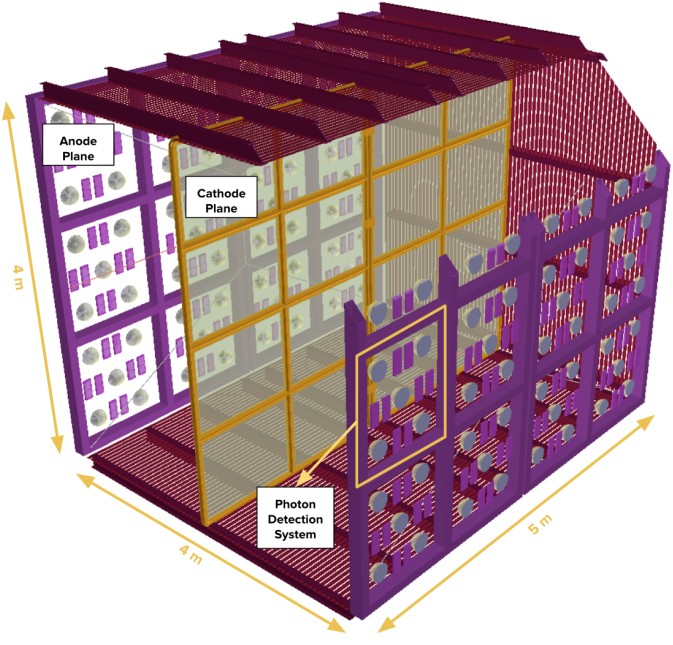

**Figure 2.** The SBND time projection chambers with photon detection system.

### 2.2. Photon Detection System

SBND has a multi-part photon detection system: 120 photomultiplier tubes and 192 X-ARAPUCA [7] devices in total sit behind both anode wire planes. Together, these devices can detect both visible and ultraviolet (the wavelength of argon's scintillation) light. The cathode plane also contains mesh panels with reflective foils, allowing for both prompt scintillation and reflected light to be detected. This leads to a high and uniform light yield throughout the detector. Light information on SBND is crucial for determining the timing of the interaction, triggering for neutrino/BSM interactions, and removing lower-light cosmic background events.

### 2.3. Cosmic Ray Tagger

There is a cosmic ray tagger made up of scintillator strips that covers all the faces of the detector with $4\pi$ coverage. The Cosmic Ray Tagger will be used to tag cosmic rays for both neutrino and BSM physics searches. In addition, it is currently being used as a "beam telescope"—see more in Section 3.4.

### 2.4. SBND PRISM

SBND's proximity to the beam target allows it to sample off-axis fluxes using a concept called SBND PRISM, similar to nuPRISM [8]. The neutrino energy spectra vary with off-axis angle, leading to reductions of various backgrounds for both neutrino oscillation signals as well as BSM phenomena. In the BNB, neutral mesons—which produce various BSM new physics phenomena—tend to be more off-axis, while charged mesons—which produce neutrinos—are focused by the beam magnetic horn to be more on-axis (see Figure 3). This feature can be harnessed for several BSM analyses on SBND.

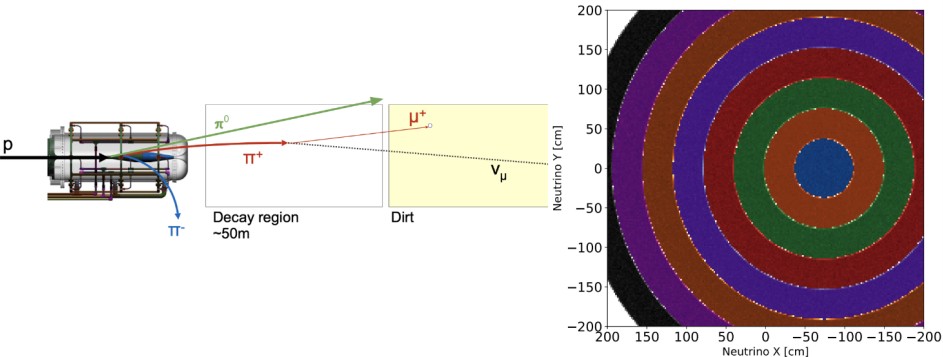

**Figure 3.** A schematic of the Booster Neutrino Beam showing charged and neutral meson production (**left**). Different off-axis angles (ranging from 0 to 1.8 degrees) for obtaining different neutrino energy spectra in SBND PRISM (**right**).

## 3. Beyond the Standard Model with SBND

In this section, we will discuss a non-exhaustive list of BSM new physics searches being developed on SBND: light dark matter, dark neutrinos, millicharged particles and heavy neutral leptons.

### 3.1. Millicharged Particles

Millicharged particles are hypothesized particles with fractional electronic charge that could be produced via neutral meson decay in the BNB [9,10]; they could be a dark matter candidate. Millicharged particles from the beam would appear in SBND as little charge blips or faint tracks that point back to the beam target. Reconstruction of low-energy blips is an active field of work, and SBND expects an energy detection threshold of 50 keV. See Figure 4.

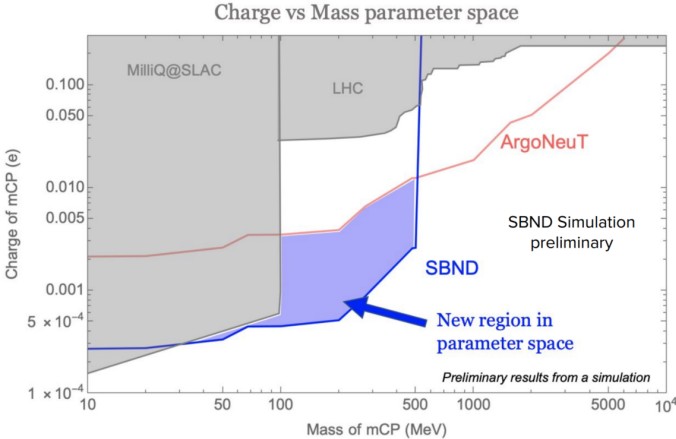

**Figure 4.** SBND's preliminary projected sensitivity in millicharged particle phase space.

### 3.2. Heavy Neutral Leptons

Heavy neutral leptons are hypothesized heavy neutrinos that are an addition to the current three-flavor paradigm [11,12]. Produced via neutral meson decay in the BNB, heavy neutral leptons would arrive later at SBND compared to neutrinos as they are heavier. Work is underway on SBND to develop a delayed trigger for these particles with respect to the neutrino beam spill, similar to the method used on MicroBooNE [13]. In addition, SBND plans to harness the excellent nanosecond-scale timing resolution of its photon detection system to look for heavy neutral leptons (and other beam-produced BSM phenomena) in between "buckets" of neutrinos within a beam spill. See Figure 5.

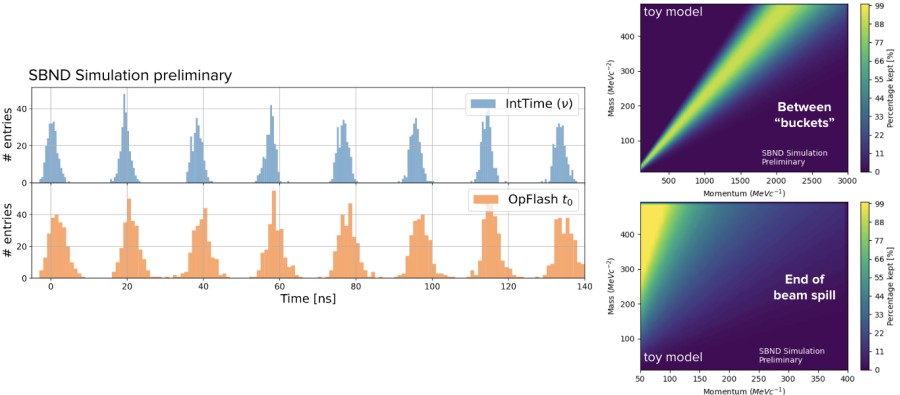

**Figure 5.** The timing structure of the BNB via light production: the light reconstruction is able to resolve the neutrino "bucket" structure of the beam spill (**left**). A preliminary estimate of different accessible areas of heavy neutral lepton phase space by SBND (**right**).

### 3.3. Light Dark Matter

Accelerator-based fixed target experiments like SBND can probe sub-GeV light dark matter postulated by "thermal relic" models, as compared to more traditional WIMP searches that are restricted to higher masses. SBND has specifically started exploring dark matter produced in vector portal light dark matter models [14], where a dark photon forms a vector mediator that kinetically mixes with the Standard Model photon, and hence can be produced via meson decay or proton bremsstrahlung in the BNB. The dark photon decays into dark matter particles, which can travel to SBND and either scatter off an electron or nucleon in the detector, or decay into electron–positron pairs. Since the typical experimental signatures of this light dark matter are a single scattered electron or an electron–positron pair, reconstructing electrons (which usually produce electromagnetic showers in a liquid argon TPC) is a key ongoing area of work on SBND. In addition, these dark matter interactions would produce no visible hadronic activity, as opposed to typical

neutrino interactions. SBND is developing tools to identify and veto hadrons, such as a proton "stub" rejection, which tags and rejects low-energy stub-like charge depositions around a vertex that fail standard track reconstruction.

### 3.4. Dark Neutrinos

Dark neutrinos are a possible BSM explanation for the low-energy excess of electromagnetic events observed by MiniBooNE [15]. They can be produced via upscattering of Standard Model neutrinos from the BNB in the dirt around SBND. These dark neutrinos are subsequently expected to decay to di-lepton pairs, which can be tagged by SBND's Cosmic Ray Tagger panels that sit upstream and downstream of the detector. Work is also underway to identify and reconstruct the decay products of dark neutrinos that enter the detector and decay inside.

## 4. Summary

SBND is a large-mass liquid argon time projection chamber on the BNB at Fermilab. Its proximity to the beam target and high-intensity neutrino beam source gives it access to large neutrino statistics and to off-axis neutrino fluxes. SBND's three-part detection subsystems—LArTPC + Photon Detection System + Cosmic Ray Tagger—allow it to reconstruct events with excellent spatial, timing and energy resolution and low energy thresholds. These features allow SBND to conduct a robust program of BSM new physics searches, and work is ongoing to estimate sensitivities for dark neutrinos, heavy neutral leptons, millicharged particles, light dark matter, and so on. The detector has been fully assembled at Fermilab, and SBND expects to begin cold commissioning in the summer of 2023.

**Funding:** This research received no external funding.

**Institutional Review Board Statement:** Not applicable.

**Informed Consent Statement:** Not applicable.

**Data Availability Statement:** Not applicable.

**Conflicts of Interest:** The author declares no conflict of interest.

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
