# Peer review of "Beyond the Standard Model New Physics Searches with SBND†"

_psf, doi:10.3390/psf8010068_

Round 1

Reviewer 1 Report

Comments and Suggestions for Authors

This is a nice, concise paper, clear and well-written.  My comments are minimal.

Comments by line number:

8.  comma after particles

10.  define SBN here

11.  define BNB here

17.  define BSM here

27.  define POT here (you use it in fig. 1)

29.  remove definition of BSM here, do it on line 17.

39.  refer to fig. 2

45.  X-ARAPUCA devices could use a reference.  (I don't know what they are.)

74. Is the detection threshold at a *mass* of 50 keV?

78. neural should be neutral

99. fail should be fails

Figures:

Can fig 3 be bigger?

Fig 4 confuses me a bit.  The simulation shows a 0.3e millicharged particle, but the plot on the right only goes up to 0.1e.  Also, the plot is in MeV, but you say in the text that you can detect particles down to masses of 50 keV.  (I think that's what you meant; please see the comment above for line 74.)  I don't see how the simulation and the text corresponds to the blue area on the plot.  Can you please clarify this in the text or the caption?
